# Differences in Socioeconomic, Dietary Choice, and Nutrition Environment Explanatory Variables for Food and Nutrition Security among Households with and without Children

**DOI:** 10.3390/nu16060883

**Published:** 2024-03-19

**Authors:** Jessica Thomson, Alicia Landry, Tameka Walls

**Affiliations:** 1USDA Agricultural Research Service, Stoneville, MS 38776, USA; tameka.walls@usda.gov; 2Department of Nutrition and Family Sciences, University of Central Arkansas, Conway, AR 72035, USA; alandry@uca.edu

**Keywords:** food security, nutrition security, dietary choice, nutrition environment, dietary behaviors, children

## Abstract

This study’s purpose was to compare socioeconomic, dietary choice, and nutrition environment variables associated with food and nutrition security in USA households with and without children. Data were collected in 2021 and consisted of households at risk of or experiencing food insecurity. Multivariable logistic regression was used to identify significant explanatory variables for food and nutrition security by household type—with or without children (<18 years of age). Food insecurity rates were 74% and 64% in households with and without children, respectively. Nutrition insecurity rates were 38% and 30% in households with and without children, respectively. For both household types, greater dietary choice increased odds (4–15 times) of food and nutrition security. In households with children, more fast-food meals increased the odds (60%) of food security, while more processed meals and greater utilization barriers to healthful meals decreased the odds (40–50%). Greater utilization barriers also decreased the odds (20%) of food security in households without children. In households with children, higher income and greater healthfulness choice increased the odds (20% and 3 times) of nutrition security, while low (vs. high) perceived limited availability of foods doubled the odds in households without children. Dietary choice is an influential and crucial factor of food and nutrition security.

## 1. Introduction

Food security has been defined as “access by all people at all times to enough food for an active, healthy life” [1]. Food insecurity, defined as low or very low food security, occurs when a household lacks money or other resources to acquire adequate food [2]. Food insecurity affects over 10% or 13 million households in the United States (US) with estimates higher for households with children (13%) and low incomes (27%), and where the head of the household has less than a high school education (25%) and is of Black race (20%) or Hispanic ethnicity (16%) [2]. Additionally, food security can impact the health of individuals and households through its relationships with dietary intake, perceptions of one’s food environment, and food that is purchased and consumed [3,4,5]. Past research has revealed differing effects of household characteristics on the relationship between food insecurity and unfulfilled medical needs between households with and without children [6]. However, whether relationships between food security and socioeconomic, dietary choice, and nutrition environment factors differ between households with and without children has yet to be determined. Such information may be useful when designing interventions aimed at improving both individual and household dietary habits.

Food security is a well-established metric for determining a household’s capability to obtain enough food. However, a recent shift in focus from food security to nutrition security has occurred. The current study uses the definition of nutrition security as “having consistent access, availability, and affordability of foods and beverages that promote well-being and prevent (and if needed, treat) disease” [7]. Many researchers and government agencies view nutrition security as the next phase for solving the nation’s disparities in nutrition quality and diet-related diseases [7]. Nonetheless, using nutrition security as a metric necessitates a careful assessment of suitable screening tools in conjunction with established measures of food security [7]. Further, similarities and dissimilarities between food and nutrition security regarding relationships with dietary choice require elucidation, including how these associations may differ between households with and without children. Therefore, the objective of the study was to identify and compare socioeconomic, dietary choice, and nutrition environment explanatory variables for food and nutrition security between households with and without children using an existing dataset.

## 2. Materials and Methods

Researchers at the Gretchen Swanson Center for Nutrition (GSCFN) collected the original data in an effort to identify food insecurity related measurement gaps and develop tools to cover these gaps [8]. GSCFN researchers wanted to investigate a more comprehensive view of food insecurity that encompassed nutritional adequacy of and personal choice in foods (e.g., identification of households able to afford food but with external limits on their capability to obtain foods that satisfy their health needs and food preferences) [8]. To test measures that were developed, individuals were recruited from five states (CA, FL, MD, NC, and WA) by partnering organizations (*n* = 7) working with households at risk of or experiencing food insecurity [8]. Recruitment occurred from April to June 2021 by means of flyers, texts, and/or emails. The inclusion criteria included ≥18 years old, understanding of English, able to answer questions about their household and themselves, and from a household at risk of or experiencing food insecurity [8]. Paper or web-based pilot surveys comprised approximately 75–85 items (subject to skip patterns) with 13 items composing the new measures. Participants received a USD 25 gift card for completing the survey (one survey per household). The University of Nebraska Medical Center Institutional Review Board reviewed and approved the study (231-20-EX). Written informed consent was provided by participants. Details of the original study are published elsewhere [8].

Household food security was assessed using the US Food Security Survey Module, consisting of 18 items for households with children <18 years of age and 10 items for households without children [9]. Categories of household food security included full (no affirmative responses), marginal (1–2 affirmative responses), low (3–7 and 3–5 affirmative responses for households with and without children, respectively), and very low (8–18 and 6–10 affirmative responses for households with and without children, respectively). Food security was dichotomized as present (full or marginal household food security) or absent/food insecurity (low or very low food security) for analytic purposes.

Household nutrition security (12-month time frame) was assessed with 4 items regarding eating foods not good for health because other types of foods could not be obtained and because healthful foods could not be obtained, concern about food they were eating hurting their health, and eating the same food several days in a row due to shortage of funds. The exact item wording is included on the GSCFN website [10]. Response options included never, rarely, sometimes, often, always, and do not know. Responses were scored from 0 to 4 points except for the do not know response (treated as missing). Mean scores were dichotomized as nutrition security (>2) or nutrition insecurity (≤2) [8]. The development, validation, and utility of the nutrition security measure are published elsewhere [8].

Perceived dietary choice was assessed with 3 items regarding a household’s capability to decide what to eat. Items covered eating foods not wanted because desired foods could not be obtained, eating food that was always changing because of uncertainty in obtaining food, and lack of control over food eaten. Perceived healthfulness choice was assessed with 3 questions regarding control for eating quality fruits and vegetables, foods good for health, and processed foods (i.e., from a box, bag, or can). Processed food examples included mac and cheese, ramen noodles, canned ravioli, and frozen TV dinners. Items covered a 12-month period. Response options included never, rarely, sometimes, often, always, and do not know. Responses were scored from 0 to 4 points except for the do not know response (treated as missing). Total choice scores were computed as means of the item scores with higher scores signifying greater choice [8]. Details about the development, validation, and utility of the dietary and healthfulness choice measures are published elsewhere [8].

Perceived limited availability was assessed with 8 items split into 2 sections. The first section included 8 types of food outlets at which food was purchased (e.g., grocery, discount, convenience, dollar). Outlet types were followed by 3 items about food availability at the outlets selected (very few quality fruits and vegetables, liked foods, and foods good for health). The second section included 4 food sources from which food was obtained (e.g., food banks, donations, grown, discarded). Food sources were followed by the same 3 items as in the first section. The second section was not used in the current analysis. Total availability scores were computed as sums of the item scores (range 0–3 points). Perceived limited availability was dichotomized as low (0–1) and high (2–3) for analytic purposes. Utilization barriers to healthful meals were assessed with 8 items regarding cooking skills (e.g., food selection or options, meal preparation, time) and storage/cooking equipment (e.g., access to cold storage, appliances, utensils, sanitation) [11]. The 3 response options were scored as 0 (never true) or 1 (sometimes true and often true). Total barrier scores were computed as sums of the item scores (range 0–8 points) with higher scores signifying more barriers. Both measures included a fourth response option, do not know, that was treated as missing in the current analysis. Questions for both measures were phrased within a 12-month period and exact question wording is included on the GSCFN website [12]. The development, validation, and utility of the perceived limited availability and utilization barriers measures are published elsewhere [11].

Sociodemographic characteristics were age (years), gender (male or female), race/ethnicity, household children <18 years of age (yes or no), and number adults in household (1 or ≥2). Socioeconomic characteristics were education, employment, annual household income (USD), and recipient of the Supplemental Nutrition Assistance Program (SNAP, government food assistance). Race/ethnicity was categorized as Black/African American, Hispanic/Latino(a), White/European American, and other race/multiracial/multiethnic (including American Indian/Alaskan Native, Asian/Asian American, Middle Eastern/North African, Native Hawaiian/Pacific Islander, and race/ethnicity not listed). Education was categorized as ≤ high school diploma [< high school or high school diploma/General Educational Development test (GED)], some college (associate degree, trade school, or professional certificate), and ≥ Bachelor’s degree (Bachelor’s degree or medical/law/graduate school). Employment categories included not working (retired, disabled, homemaker, full-time student), working <30 h/week, and working ≥30 h/week. The annual household income ratio was calculated by multiplying the median of the selected range (14 discrete ranges) by 12 months divided by number in household/1000 (USD 1000 per household member).

SAS^®^, version 9.4 (SAS Institute Inc., Cary, NC, USA) was used for the statistical analyses with *p* ≤ 0.05 as the significance level. Participants who had complete food and nutrition security data and indicated how many adults were in the household were included in the current analysis. Participant characteristics were summarized using frequencies, percentages, means, and standard deviations (SDs). Comparisons between participants categorized by presence/absence of child(ren) <18 years of age (households with children vs. households without children) were performed using chi square tests of association for categorical variables and Wilcoxon rank-sum 2-sample tests for ordinal and continuous variables.

Multivariable logistic regression was used to model the probabilities of having food and nutrition security separately and to calculate odds ratios and their 95% Wald confidence intervals for significant effects (explanatory variables) for households with and without children. Explanatory variables included age; education; employment; income ratio; fruit and vegetable intake; scratch-cooked, fast-food, and processed meals consumption; dietary and healthfulness choice; utilization barriers; perceived limited availability; and shopping at dollar and convenience stores. The presence of multicollinearity was assessed based on condition indices (>10) and variance inflation factors (>5) [13]. All condition indices were ≤3, while all variance inflation factors were <2, indicating multicollinearity was not present. Variance estimation was computed using Taylor series and missing values were treated as not missing completely at random. Explanatory variables were iteratively removed from the models until only significant variables were retained. The discriminative ability of the logistic regression models was assessed based on concordance (%).

## 3. Results

Of the 486 participants in the original dataset, 427 (88%) were included in the current analysis. Characteristics of the analytic sample by household type are presented in Table 1. In total, 60% percent of households had at least one child <18 years of age and 40% of households did not have any children. Comparisons between household types revealed several differences. Households with children were more likely to have at least two residing adults (46% vs. 35%), receive SNAP benefits (66% vs. 48%), and have food insecurity (74% vs. 64%) as compared to households without children. Additionally, participants from households with children were more likely to be female (86% vs. 63%), be younger (39 vs. 54 years of age), and have a lower income [3.7 vs. 10.0 (USD 1000 per household member)] and less education (14% vs. 26% with ≥ a Bachelor’s degree) than participants from households without children. For nutrition environment and dietary choice, participants from households with children were more likely to shop at dollar stores (60% vs. 41%) and convenience stores (35% vs. 25%), perceive limited food availability as high (45% vs. 32%), consume more fast-food (1.0 vs. 0.9 times/week) and processed meals (2.1 vs. 1.4 times/week), and have lower healthfulness choice (2.4 vs. 2.6) than participants from households without children.

### 3.1. Food Security

Multivariable logistic regression results for food security by household type are presented in Table 2. For households with children, the consumption of fast food and processed meals, dietary choice, and utilization barriers were significant explanatory variables for food security. For every 1 day/week increase in the consumption of fast-food meals, the odds of having food security increased by 60%. Conversely, for every 1 day/week increase in the consumption of processed meals, the odds of having food security decreased by 40%. For every one-point increase in dietary choice, the odds of having food security increased 8-fold. In contrast, for every one-point increase in utilization barriers, the odds of having food security decreased by 50%. The model concordance was 93%.

For households without children, dietary choice and utilization barriers were significant explanatory variables for food security. For every one-point increase in dietary choice, the odds of having food security increased 6-fold. Conversely, for every one-point increase in utilization barriers, the odds of having food security decreased by 20%. The model concordance was 87%.

### 3.2. Nutrition Security

Multivariable logistic regression results for nutrition security by household type are presented in Table 2. For households with children, dietary choice healthfulness choice, and income ratio were significant explanatory variables for nutrition security. For every one-point increase in dietary and healthfulness choice, the odds of having nutrition security increased 15- and 3-fold, respectively. For every USD 1000 increase in income per household member, the odds of having nutrition security increased by 20%. The model concordance was 91%.

For households without children, dietary choice and perceived limited availability were significant explanatory variables for nutrition security. For every one-point increase in dietary choice, the odds of nutrition security increased 4-fold. Participants with low perceived limited availability had five times the odds of having nutrition security than participants with high perceived limited availability. The model concordance was 84%.

## 4. Discussion

In this study, socioeconomic, dietary choice, and nutrition environment explanatory variables for food and nutrition security were identified and compared between households with and without children. While several factors differentially explained food and nutrition security among households with and without children, only dietary choice was a significant factor for both food and nutrition security and for both household types. The association between food security and dietary choice is logical because households with food insecurity have limited financial resources that would restrict their food choices, thus affecting control over the types of foods they consume. Likewise, the association between dietary choice and nutrition security is reasonable because households with food insecurity are probably experiencing nutrition insecurity as well [3]. Constrained financial resources of households with food insecurity may make it difficult to obtain nutritionally balanced foods thus negatively impacting nutrition security [3]. While the associations observed between food and nutrition security and dietary choice were present for both household types, the observed effects were larger for households with children. These results suggest that households with children who have limited or no control over their dietary choices are more likely to have food and nutrition insecurity than households without children who experience the same limitations in dietary choice. Thus, policies designed to improve nutrition environments of disadvantaged or marginalized communities may increase food and nutrition security in all household types with a potentially larger impact on households with children.

While dietary choice explained both food and nutrition security in both household types, healthfulness choice was a significant factor only for nutrition security and only in households with children. The positive relationship between healthfulness choice and nutrition security suggests that having greater choice in foods that are believed to be good for health (e.g., fruits and vegetables) results in the consumption of these foods and, thus, protects against nutrition insecurity. It is intriguing that the relationship was not observed in households without children. Adult food preferences are considered relatively stable, as evidenced by difficulties in making long-lasting healthful changes in one’s diet [14]. Hence, whether adults believe they have greater choice and control over healthful foods may have little effect on their nutrition security because they are not changing their dietary habits. However, adults who are parents may be more conscious of and willing to purchase healthful foods for their children when such foods are available. In a study conducted using Nielson Homescan Consumer Panel data, the authors found that produce purchases increased after the initiation of parenthood, although the effect was driven by households with higher incomes [15]. It also is possible that children are influencing their parents’ food purchases. In a study conducted with grade school children and their parents, the authors found that children were more likely to request fruits and vegetables when they accompanied their parents during grocery shopping trips [16]. Taken together, these results suggest that children directly and indirectly affect their parents’ healthful food purchases, potentially leading to a higher likelihood of nutrition security in the household. Researchers should consider taking advantage of parents’ willingness to purchase more healthful foods for their children and their children’s potentially positive effects on parents’ food purchases when designing interventions to improve the nutritional status of families.

The positive association observed between processed meals consumption and food security supports evidence previously reported in the literature. In a nationally representative sample of US adults, more severe food insecurity was associated with higher intakes of ultra-processed foods [17]. Similarly, in a nationally representative sample of Canadian children and adults, percent energy from ultra-processed foods was associated with the severity of food insecurity [18]. However, the positive association observed between fast-food meal consumption and food security is surprising and contradicts results previously reported in the literature. Researchers have observed either no relationship between the frequency of fast-food consumption and food security [19,20] or a negative relationship (food insecurity associated with fast-food consumption but only in lower- and middle-income countries) [21]. Clearly, the relationship between food security and fast-food consumption deserves further study.

The finding that fewer utilization barriers was protective against food insecurity but not nutrition insecurity is puzzling. It may be that the utilization barriers instrument used in the current study was more sensitive to food security/insecurity than nutrition security/insecurity, given that five of the eight items addressed general food barriers, while three addressed healthful foods/meals in particular. It also is possible that other factors explained most of the variability in nutrition security that was associated with utilization barriers, thus making its effect nonsignificant. Others have reported relationships between barriers to healthful food preparation and food insecurity. Based on parent data from an elementary school nutrition intervention, researchers found that households with food insecurity were more likely to report barriers to preparing and cooking vegetables than households with food security [22]. In a study conducted with Native Americans with type 2 diabetes, participants identified factors contributing to food insecurity beyond food cost, including preparation time for and limited cooking knowledge of fresh produce [23]. Thus, interventions that are designed to improve knowledge of—and cooking skills for—healthful foods may increase cooking confidence, leading to improved dietary intakes in households [24,25].

In contrast to utilization barriers, the perception of low limited availability of foods was protective against nutrition insecurity but not food insecurity. A possible explanation is that two of the three items for the instrument specifically addressed healthful foods, while the third item addressed preferred foods. Hence, the perceived limited availability tool may be more relevant to nutrition security than food security. Counter to the current study, associations between food insecurity and limited availability (less likely to report a varied selection of high-quality fruits and vegetables in a neighborhood) were reported in a study conducted among households with children [4]. Additionally, other researchers have reported a greater likelihood of food security among individuals who reported less difficulty accessing fruits and vegetables and better-quality stores in their neighborhoods than their counterparts reporting more difficulty and lower-quality stores [26]. These seemingly disparate findings indicate that more work is needed to elucidate the relationships between perceptions of the nutrition environment and food and nutrition security.

The deliberate sampling of populations at risk of or experiencing food insecurity, leading to a majority-low-income sample, may explain the lack of effect of annual household income on food security. However, the observed effect of income on nutrition security for households with children suggests that even among households with low incomes, relatively small increases in income provide some protection against nutrition insecurity. Thus, interventions or policies designed to increase the earning capacity of parents with low income (e.g., vocational education, on-the-job training) may lead to increased nutrition security, and thus, improved dietary intake for households with children.

The study limitations include the restricted generalizability of the study findings due to the use of a convenience sample. The inclusion of only individuals at risk of or experiencing food insecurity may have limited associative findings between food and nutrition security and explanatory variables. All data are self-reported and, therefore, subject to bias.

## 5. Conclusions

Dietary choice appears to be an influential and crucial factor of food and nutrition security in households at risk of or experiencing food insecurity. Healthfulness choice seems to play an important role in nutrition security for households with children, while perceived limited availability seems more relevant to households without children. Thus, the presence of children in households may need to be considered when designing interventions or proposing policy to reduce food and nutrition insecurity.

## Figures and Tables

**Table 1 nutrients-16-00883-t001:** Participant characteristics by presence of child(ren) in household (*n* = 427) ^a^.

	≥1 Child		No Child		
	(*n* = 255, 60%)		(*n* = 172, 40%)		
Characteristic	*n*	%	*n*	%	*p* ^b^
≥2 adults (household)	193	75.7	86	50.0	<0.001
Female gender	214	85.6	107	63.3	<0.001
Race					<0.001
Black/African American	55	22.2	19	11.5	
Hispanic/Latino(a)	77	31.1	23	13.9	
White/European American	82	33.1	99	60.0	
Other race/multiracial/multiethnic ^c^	34	13.7	24	14.5	
Education					0.008
≤ High school diploma ^d^	114	45.8	64	39.3	
Some college ^e^	100	40.2	59	36.2	
≥ Bachelor’s degree	35	14.1	43	26.4	
Employment					0.004
Not working ^f^	143	57.4	122	73.1	
Working <30 h/week	60	24.1	22	13.2	
Working ≥30 h/week	46	18.5	23	13.8	
SNAP recipient	167	65.5	83	48.3	<0.001
Food insecurity ^g^	189	74.1	110	64.0	0.025
Nutrition insecurity ^h^	97	38.0	51	29.7	0.074
Shop dollar stores	152	59.6	70	40.7	<0.001
Shop convenience/corner stores	89	34.9	43	25.0	0.030
Perceived limited availability					0.004
None	38	18.4	48	34.5	
Low	36	17.5	26	18.7	
Moderate	39	18.9	21	15.1	
High	93	45.1	44	31.7	
	Mean	SD	Mean	SD	*p* ^i^
Age (years)	39.3	10.5	54.1	15.1	<0.001
Annual income ratio ^j^	3.7	3	10.0	7	<0.001
No. individuals in household	5.3	2.9	1.7	1	<0.001
FV intake (4 types/day)	2.0	1.5	2.0	1.5	0.576
Scratch cooked meals (days/week)	3.3	2.2	3.2	2.5	0.641
Fast-food meals (days/week)	1.0	1.1	0.9	1.3	0.021
Processed/heated meals (days/week)	2.1	1.8	1.4	1.7	<0.001
Healthfulness choice (mean score) ^k^	2.4	0.9	2.6	0.9	0.004
Dietary choice (mean score) ^k^	2.5	0.9	2.6	0.9	0.145
Utilization barriers (sum score) ^l^	2.3	2.2	2.2	2.4	0.470

SD, standard deviation; FV, fruit and vegetable; SNAP, Supplemental Nutrition Assistance Program. ^a^ child(ren) <18 years of age. ^b^ *p*-value for chi square tests. ^c^ includes American Indian/Alaskan Native, Asian/Asian American, Middle Eastern/North African, Native Hawaiian/Pacific Islander, and race/ethnicity not listed. ^d^ includes General Educational Development test (GED). ^e^ includes associate degree, trade school, or professional certificate. ^f^ includes retired, disabled, homemaker, or full-time student. ^g^ food insecurity = low or very low; food security = full or marginal. ^h^ nutrition insecurity = score ≤2; nutrition security = score >2 (range 0–4 points). ^i^ *p*-value for Wilcoxon 2-sample tests. ^j^ USD 1000 per household member. ^k^ higher values indicate greater choice (range 0–4 points). ^l^ higher values indicate more perceived barriers (range 0–8 points).

**Table 2 nutrients-16-00883-t002:** Multivariable logistic regression models for food and nutrition security by presence of child(ren) in household ^a,b^.

	≥1 Child				No Child			
Characteristic	OR	95% CI		*p*	OR	95% CI		*p*
	*Food Security*							
Dietary choice	7.7	3.7	15.7	<0.001	6.4	3.1	13.0	<0.001
Processed meals (days/week)	0.6	0.4	0.8	0.003	NS	-	-	-
Fast-food meals (days/week)	1.6	1.1	2.5	0.020	NS	-	-	-
Utilization barriers	0.5	0.4	0.8	0.001	0.8	0.6	1.0	0.047
	*Nutrition Security*							
Dietary choice	14.6	6.6	32.6	<0.001	4.1	2.2	7.7	<0.001
Healthfulness choice	2.7	1.5	4.8	0.001	NS	-	-	-
Perceived limited availability ^c^	NS	-	-	-	5.4	2.0	14.5	0.001
Annual income ratio ^d^	1.2	1.0	1.3	0.018	NS	-	-	-

OR, odds ratio; CI, confidence interval; NS, not significant at 0.05 level. ^a^ children <18 years of age. ^b^ variables included in models = age; gender; education; employment; income ratio; fruit and vegetable intake; scratch-cooked, fast-food, and processed meals consumption; dietary and healthfulness choice; utilization barriers; perceived limited availability; and shopping at dollar and convenience stores. ^c^ high vs. low. ^d^ USD 1000 per household member.

## Data Availability

Data were obtained from the Gretchen Swanson Center for Nutrition (https://www.centerfornutrition.org/food-security-measures, accessed on 16 May 2023) and are available with the permission of Dr. Eric Calloway, Senior Research Scientist.

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
