# Peer review of "Differences in Socioeconomic, Dietary Choice, and Nutrition Environment Explanatory Variables for Food and Nutrition Security among Households with and without Children"

_nutrients, 2024, doi:10.3390/nu16060883_

Round 1
Reviewer 1 Report
Comments and Suggestions for Authors
The authors aim is to compare socio-economic, food choices, and nutrition environment in households with and without children, associating it with food and nutrition security.
the objective and results of the study are interesting. However, some parts must be improved:
1. Introduction: the definition of food insecurity is a citation or an elaboration of the authors?
2. Results: the percentage results are sometimes written in full (e.g. sixty) and sometimes in number (e.g. 40%). Please, always use the same way
3. The structure of the text is occasionally difficult to comprehend and is not consistently expressed linearly. Certain sections ought to be rewritten more cohesively.
Comments on the Quality of English Language
As said before, some sections need to be revise in terms of linearity
Author Response
We thank the reviewers for their thoughtful review of our manuscript. We have incorporated most of the suggestions provided and believe that the resulting revisions have improved the clarity of our manuscript. We hope our responses and manuscript edits (red font) have adequately addressed the reviewers’ concerns. Our responses to the reviewers’ comments and suggestions follow.
Reviewer 1
- Does the introduction provide sufficient background and include all relevant references? [Can be improved.]
Response: Please see our response to comment #8.
- Are all the cited references relevant to the research? [Yes]
Response: Thank you for acknowledging the relevancy of our references.
- Is the research design appropriate? [Yes]
Response: Thank you for acknowledging the appropriateness of our research design.
- Are the methods adequately described? [Yes.]
Response: Thank you for your positive review of our methods.
- Are the results clearly presented? [Can be improved.]
Response: Please see our response to comment #9.
- Are the conclusions supported by the results? [Can be improved]
Response: Please see our response to comment #10.
- The authors’ aim is to compare socio-economic, food choices, and nutrition environment in households with and without children, associating it with food and nutrition security. The objective and results of the study are interesting. However, some parts must be improved:
Response: Thank you for acknowledging the usefulness of our research. We have addressed your concerns in the items that follow.
- Introduction: the definition of food insecurity is a citation or an elaboration of the authors?
Response: We apologize for our missing reference which has now been added (line 30).
- Results: the percentage results are sometimes written in full (e.g. sixty) and sometimes in number (e.g. 40%). Please, always use the same way.
Response: Thank you for this suggestion. It is standard practice to write out numbers when they begin a sentence as we did for the 60% (line 169). This is the only instance in which we started a sentence with a number.
- The structure of the text is occasionally difficult to comprehend and is not consistently expressed linearly. Certain sections ought to be rewritten more cohesively.
Response: Thank you for this suggestion. Our introduction begins with food security and progresses to nutrition security. Our methods begin by describing the original study, then describing food and nutrition security measures (outcomes), followed by the explanatory variables (dietary and healthfulness choice, perceived limited availability, utilization barriers, and sociodemographic/socioeconomic characteristics), and concluding with the statistical analysis methods. Our results begin with a description of the sample and comparisons between households with and without children, followed by results from the food security models, and then the nutrition security models. Finally, our discussion begins with similarities between the food and nutrition security model explanatory variables (dietary choice), followed by differences in model explanatory variables. Thus, we feel that the flow/progression is consistent and logical between the sections. However, we have rearranged the explanatory variables in Table 2 to reflect the order presented in the discussion. We hope this reorganization addresses your concern about linear flow.
Reviewer 2 Report
Comments and Suggestions for Authors
This is an interesting paper on nutritional issues of high social interest. A few queries and suggestions. Data were collected in five States, but no further details are offered. What States? Any differences? I believe that data of Table 1 (a long list of indigestible figures) would be more clearly understandable if they were shown as a series of separate graphs, possibly pie charts. Part 3 (results): What were the reasons that led the authors to exclude 12% of participants from the study?
Author Response
We thank the reviewers for their thoughtful review of our manuscript. We have incorporated most of the suggestions provided and believe that the resulting revisions have improved the clarity of our manuscript. We hope our responses and manuscript edits (red font) have adequately addressed the reviewers’ concerns. Our responses to the reviewers’ comments and suggestions follow.
Reviewer 2
- Does the introduction provide sufficient background and include all relevant references? [Yes]
Response: Thank you for your positive review of our introduction.
- Are all the cited references relevant to the research? [Yes]
Response: Thank you for acknowledging the relevancy of our references.
- Is the research design appropriate? [Yes]
Response: Thank you for acknowledging the appropriateness of our research design.
- Are the methods adequately described? [Can be improved.]
Response: Please see our response to comment #8.
- Are the results clearly presented? [Can be improved.]
Response: Please see our responses to comments #9 and #10.
- Are the conclusions supported by the results? [Yes]
Response: Thank you for your positive review of our conclusions.
- This is an interesting paper on nutritional issues of high social interest. A few queries and suggestions.
Response: Thank you for acknowledging the value of our paper. We have addressed your queries and suggestions in the items that follow.
- Data were collected in five States, but no further details are offered. What States? Any differences?
Response: Thank you for this suggestion. We have added the state names to the methods (line 67). Conducting analyses by each state was beyond the scope of this manuscript. Additionally, small sample sizes would negate confidence in results analyzed by state (e.g., some states had n<10 for participants with nutrition security/insecurity in households with or without children).
- I believe that data of Table 1 (a long list of indigestible figures) would be more clearly understandable if they were shown as a series of separate graphs, possibly pie charts.
Response: Thank you for this suggestion. We understand that Table 1 is lengthy and hence have chosen to collapse some of the categories [adults (household) and race] and report only one category for dichotomous characteristics [gender, SNAP recipient, food security (reporting food insecurity), nutrition security (reporting nutrition insecurity), shop dollar stores, and shop convenience stores]. We also removed the 4 categories for food security as it was somewhat redundant with the food security/insecurity characteristic. We hope the shortened table sufficiently addresses your concerns.
- Part 3 (results): What were the reasons that led the authors to exclude 12% of participants from the study?
Response: Thank you for this question. We included only participants who had complete food and nutrition security data and indicated how many adults were in the household (lines 144-146).
Reviewer 3 Report
Comments and Suggestions for Authors
The abstract must indicate that the data is from the USA.
Household food security: It is understood according to the document that the instrument used (questionnaire) was different between households with children and without children. The authors should comment on that and its effects on the results. It is also advisable to present the scales used or leave a link or cite where these scales can be seen.
The authors must include the hit (success, classification accuracy) rate of each of their logit regressions.
Author Response
We thank the reviewers for their thoughtful review of our manuscript. We have incorporated most of the suggestions provided and believe that the resulting revisions have improved the clarity of our manuscript. We hope our responses and manuscript edits (red font) have adequately addressed the reviewers’ concerns. Our responses to the reviewers’ comments and suggestions follow.
Reviewer 3
- Does the introduction provide sufficient background and include all relevant references? [Can be improved.]
Response: Please see our response to comment #7.
- Are all the cited references relevant to the research? [Must be improved.]
Response: Please see our response to comment #8.
- Is the research design appropriate? [Can be improved.]
Response: Please see our response to comment #8.
- Are the methods adequately described? [Can be improved.]
Response: Please see our response to comment #8.
- Are the results clearly presented? [Must be improved.]
Response: Please see our response to comment #9.
- Are the conclusions supported by the results? [Must be improved.]
Response: We have reviewed our conclusions and do not see any that we feel are not supported by the results. If you could identify the conclusions that you feel are not supported, we would be happy to review and modify them accordingly.
- The abstract must indicate that the data is from the USA.
Response: Thank you for this suggestion. We have included “USA” in our abstract (line 10).
- Household food security: It is understood according to the document that the instrument used (questionnaire) was different between households with children and without children. The authors should comment on that and its effects on the results. It is also advisable to present the scales used or leave a link or cite where these scales can be seen.
Response: Thank you for these suggestions. We did not use different measures to classify food security in households with and without children. Food security was measured using the established and accepted US Food Security Survey Module. The same survey was used for all participants. If a household did not contain children <18 years of age, then certain questions were not relevant to that household and thus they were not asked the questions. It’s the same concept used in many survey instruments. For example, in the NIH fruit and vegetable screener, the first question asks if a participant drank 100% fruit juice. If they answer no, then they skip the question concerning frequency of intake because it is not relevant to them. We did reference the US Food Security Survey Module (ref #9) and it does contain a link (URL) at which the survey may be found and downloaded.
- The authors must include the hit (success, classification accuracy) rate of each of their logit regressions.
Response: Thank you for this suggestion. We have added the % concordant for each of the logistic regression models to our results and added the measure to the methods (lines 164-165, 194, 198-199, 206-207, 212).